# Transverse sinus injections drive robust whole-brain expression of transgenes

Ali S Hamodi[1]*, Aude Martinez Sabino[1,2], N Dalton Fitzgerald[1], Dionysia Moschou[1], Michael C Crair[1,3,4]*

[1]Department of Neuroscience, Yale School of Medicine, New Haven, United States; [2]University of Technology of Compiègne, Compiègne, France; [3]Department of Ophthalmology and Visual Science, Yale School of Medicine, New Haven, United States; [4]Kavli Institute for Neuroscience, Yale School of Medicine, New Haven, United States

**Abstract** Convenient, efficient and fast whole-brain delivery of transgenes presents a persistent experimental challenge in neuroscience. Recent advances demonstrate whole-brain gene delivery by retro-orbital injection of virus, but slow and sparse expression and the large injection volumes required make this approach cumbersome, especially for developmental studies. We developed a novel method for efficient gene delivery across the central nervous system in neonatal mice and rats starting as early as P1 and persisting into adulthood. The method employs transverse sinus injections of 2–4 µL of AAV9 at P0. Here, we describe how to use this method to label and/or genetically manipulate cells in the neonatal rat and mouse brain. The protocol is fast, simple, can be readily adopted by any laboratory, and utilizes the widely available AAV9 capsid. The procedure is adaptable for diverse experimental applications ranging from biochemistry, anatomical and functional mapping, gene expression, silencing, and editing.

## Introduction

Recombinant adeno-associated viruses (AAVs) are commonly used vectors for *in vivo* gene delivery (*Foust et al., 2009*), and recent work demonstrates whole-brain gene delivery by retro-orbital injection of AAV9 and other engineered AAV variants (*Chan et al., 2017*). However, the onset of robust systemic expression of genes through AAVs traditionally occurs several weeks after the time of injection (*Chan et al., 2017*). This presents a particular challenge for experiments requiring gene manipulation early in postnatal development. In addition, current gene-delivery methods, such as retro-orbital injections, are difficult and particularly disruptive in young neonates. Other options, such as intravenous administration of virus through the tail or temporal vein, require excessively high injection volumes (100 µL) that are expensive and disruptive, and still produce inefficient transduction (15–18%) of target cells (*Foust et al., 2009*). Intracerebroventricular (ICV) injections can yield strong expression in neonatal brains using small quantities of virus, but results in non-uniform expression across the cortex and requires penetrating both brain hemispheres with a 32-gauge needle, which leads to substantial damage to the cortex and cell death (*Kim et al., 2013*; *McLean et al., 2014*; *Kim et al., 2014*; *Passini and Wolfe, 2001*). Traditional alternatives, such as targeted gene expression through the creation of transgenic mice, are limited by the burden of interbreeding, which typically necessitates complex and time-consuming breeding schemes to drive conditional expression in desired cell types (*Madisen et al., 2015*; *Daigle et al., 2018*).

To remedy these limitations, we developed an easy and efficient method of transgene delivery through the transverse sinus in neonatal mice, which we refer to as 'n-SIM' (neonatal sinus injection method). The transverse sinus is easily accessible along the posterior edge of the forebrain and convenient for virus injection in neonates when the skin and skull remain quite thin. AAV9 is very well

**\*For correspondence:**
ali.hamodi@yale.edu (ASH);
michael.crair@yale.edu (MCC)

**Competing interests:** The authors declare that no competing interests exist.

suited for this approach given the proclivity of AAV9 to cross the blood-brain barrier (BBB), especially in young animals (*Foust et al., 2009*). The position and proximity of the transverse sinus to the brain also makes it particularly well suited for gene delivery to the brain. Indeed, transverse sinus injection of as little as 2–4 µL of AAV9 ($1 \times 10^{13}$ vg/mL) into neonatal mice results in robust and widespread gene delivery to the brain. With virus injection at P0, we observe dense labeling in cortex, thalamus, midbrain, and hippocampus as early as P4-P5 that persists into adulthood. Sinus injections with AAV9 were successfully tested in both mice and rats but are likely suitable for any mammalian species.

This method enables the targeting of distinct cell populations at early stages of development and permits the delivery of multiple viral constructs at the same time across the whole brain. n-SIM will dramatically accelerate the application of novel molecular technologies without the need to generate costly transgenic strains or generate complicated crosses. Sinus injections also circumvent the caveats of direct injections into the brain parenchyma, which can cause tissue damage and variable gene expression. This is especially important for neurodevelopmental studies, but the approach is also applicable to older animals as the expression persists in adults. Our method provides an easy and fast way (10 min per pup) to express a wide variety of transgenes across the extent of the brain using the easily accessible AAV9 capsid. Finally, sinus injected animals show no deleterious health effects, either from the injections or as a result of high expression levels typically associated with early embryonic expression in transgenic mice (*Daigle et al., 2018*). We describe below in detail how our method is performed and demonstrate its utility in carrying out experiments in neonatal mice and rats to answer fundamental questions that were previously impractical or impossible.

## Results

### AAV9 n-SIM yields robust whole-brain expression of transgenes during the first postnatal week

Transduction of neurons in the mouse brain following neonatal injection of AAV9 in the transverse sinus ('n-SIM') was robust and widespread. Injection of 4 µL of AAV9 expressing GCaMP6 under the control of the synapsin promoter (AAV9-syn-GCaMP6s) at P0-P1 (*Figure 1A*) resulted in widespread expression of GCaMP6 across the cortical mantle at P5 (*Figure 1B,C*) that persisted into adulthood (*Figure 1D–F*). With n-SIM, we observed labeling of 52 +/- 12%, 50 +/- 6%, 51 +/- 14%, and 70 +/- 7% of cortical neurons at P6, P9, P14, P45 respectively (*Figure 1H*, *Figure 1—source data 1*; P6, n = 5; P9, n = 2; P14, n = 5; P45, n = 5). Expression was robust in all cortical regions examined (*Figure 1I,J–M*; *Figure 1—source data 2*; P14: M1: 47 +/- 15%, n = 5; V1: 63 +/- 6%, n = 4; S1: 61 +/- 8%, n = 3; A1: 58 +/- 7%, n = 2; Retrosplenial: 58 +/- 4, n = 2; Piriform: 44 +/- 17%, n = 3), and across all cortical layers (*Figure 1N*, *Figure 1—source data 3*; P14, layer 2/3: 52 +/- 17%; layer 4: 63 +/- 13%, layer 5: 61 +/- 12%, layer 6: 53 +/- 17%; n = 5). n-SIM was also effective at transduction of GCaMP6 in subcortical areas, with 66 +/- 16%, 49 +/- 5%, 54 +/- 4%, and 85 +/- 15% of thalamic neurons displaying expression at P6, P9, P14, and P45 respectively (*Figure 1H*, *Figure 1—source data 2*; P6, n = 2, P9, n = 2; P14, n = 4), irrespective of thalamic subregion (*Figure 1I*, *Figure 1—source data 2*; O-R; P14: DLG: 52 +/- 2%; n = 2, Anterodorsal thalamic nucleus: 60%, n = 2). Expression was also robust in hippocampus (*Figure 1I,T–W*, *Figure 1—source data 2*; P14: 43 +/- 28%, n = 3) and superior colliculus (*Figure 1I*, *Figure 1—source data 2*; 66 +/- 2%, n = 2), though expression in striatum was somewhat less ubiquitous (*Figure 1I*, *Figure 1—source data 2*; 16 +/- 16%, n = 2).

Overall, these results compare favorably to the sparse transfection levels reported previously using temporal vein injections of AAV9 at P1. For example, Foust et al. observed a cellular transfection rate of 15% and 18% in cortex at P11 and P20, respectively, after temporal vein injection at P1 (*Foust et al., 2009*). However, Foust et al. used an AAV9 vector that expresses GFP under the control of chicken B-actin hybrid (CB) promoter. For a more direct comparison of the efficacy of transfection with temporal vein and transverse sinus injections, we injected the temporal vein at P1 with the same AAV9 construct used for transverse sinus injections (AAV9 expressing GCaMP6s under the synapsin promoter) and measured expression at P14 and P21. We observed much weaker, slower, and non-uniform GCaMP expression relative to AAV9 delivery through the transverse sinuses

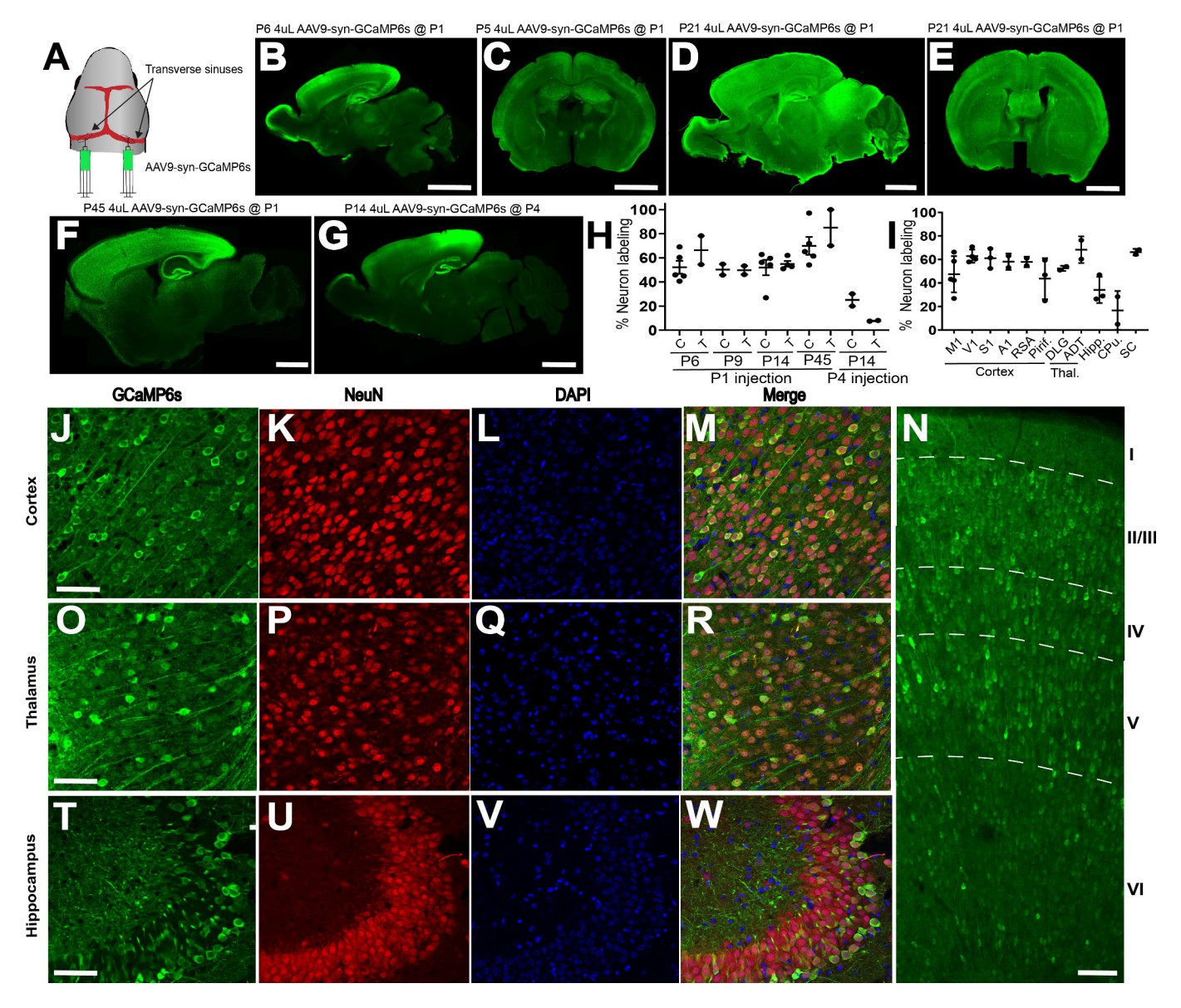

**Figure 1.** AAV9 n-SIM at P0-P1, but not at P4, leads to widespread neuronal transduction in the neonatal mouse brain. (**A**) Schematic showing sites of viral injection at P1. (**B,C**) Example sagittal and coronal sections of P6 and P5 mouse brains, respectively, showing widespread expression of GCaMP6s across the cortex and several other brain regions including hippocampus, midbrain, and thalamus. Scale bar = 2 mm. Exposure time: 1000 ms. Minimum-Maximum display range in ImageJ (Unsigned 16-bit range): (**B**) 73–1200; (**C**) 0–4095. (**D–E**) Example sagittal and coronal sections at P21 showing that expression of GCaMP6s at P21 is brighter than at earlier ages. Exposure time: 1000 ms. Minimum-Maximum display range in ImageJ (Unsigned 16-bit range)=0–4095. (**F**) Sagittal section of P45 brain injected at P1 with 4 µL of AAV9 ($1 \times 10^{13}$ vg/mL. Exposure time: 500 ms. Minimum-Maximum display range in ImageJ (Unsigned 16-bit range)=73–3500. G: Sagittal section of P21 brain injected at P4 with 4 µL of AAV9 ($1 \times 10^{13}$ vg/mL. Exposure time: 1000 ms. Minimum-Maximum display range in ImageJ (Unsigned 16-bit range)=73–2000. (**H**) Quantification of cortical and thalamic neuron labeling at P6, P9, and P14 after P1 (P6, n = 5; P9, n = 2; P14, n = 5; P45, n = 5) or P4 (P14, n = 2) sinus injections. Each data point in the plot represents an individual brain. Horizontal lines represent the mean, and vertical lines represent the standard deviation. C = cortex, T = thalamus. (**I**). Quantification of neuron labeling at P14 in different cortical and thalamic regions, in addition to hippocampus and striatum. M1 = motor cortex (n = 5), V1 = visual cortex (n = 4), S1 = somatosensory cortex (n = 3), A1 = auditory cortex (n = 2), RSA = retrosplenial cortex (n = 2), Piri = piriform cortex (n = 3), DLG = dorsolateral geniculate nucleus (n = 2), ADT = anterodorsal thalamic nucleus (n = 1), Hipp = hippocampus (n = 3), CPu = caudate and putamen (n = 2), SC = superior colliculus (n = 2). Confocal images showing GCaMP6s expression in mouse cortex and thalamus (**J–W**) at P14 after transverse sinus injection of 4 µL of $1 \times 10^{13}$ vg/mL AAV9-syn-GCaMP6s at P1. Panels (**J**), (**O**), (**T**) show abundant GCaMP6s expression in cortex, thalamus, and hippocampus, and localization with both NeuN and DAPI (**M,R,W**). Scale bar = 20 µm. (**N**). Confocal image of GCaMP6s revealing dense and widespread expression across all cortical layers at P14. Scale bar is 40 µm.

*Figure 1 continued on next page*

*Figure 1 continued*

The online version of this article includes the following source data and figure supplement(s) for figure 1:

**Source data 1.** Quantification of neuronal labeling achieved through different injection methods.
**Source data 2.** n-SIM neuronal labeling in different brain regions.
**Source data 3.** n-SIM neuronal labeling in cortical layers.
**Source data 4.** Quantification of neuronal labeling at different ages.
**Figure supplement 1.** Comparison of AAV9 n-SIM to other methods.
**Figure supplement 2.** Neonatal transverse sinus injection method.

(*Figure 1—figure supplement 1A,B,G*, *Figure 1—source data 1*; AAV9 temporal vein injection: 9 +/- 3%, n = 4; AAV9 n-SIM: 52 +/- 4%, n = 10. AAV9 temporal injection vs. AAV9 n-SIM: p<0.0001).

n-SIM was most effective with transverse sinus injections performed just after birth. Injections performed at P4 resulted in significantly lower expression than injections at P0-P1, when measured at P14 (*Figure 1G,H*; *Figure 1—source data 4*; P14, cortex: 25 +/- 7%; thalamus: 8 +/- 0%, n = 2. AAV9 n-SIM at P1 vs. AAV9 n-SIM at P4: p=0.059 for cortex, p<0.0001 for thalamus), likely due to the rapid maturation of the BBB soon after birth. n-SIM with AAV9 was also much more efficacious than other serotypes, including AAV-PHP.eB, AAV5, and AAV1. All three alternative serotypes yielded relatively sparse expression in comparison to AAV9, even when using up to double the volume (8 µL) of virus, and expression was generally limited to superficial cortical layers (*Figure 1—figure supplement 1C–G*, *Figure 1—source data 1*; AAV5 n-SIM: 4 +/- 2%, n = 4; AAV1 n-SIM: 33 +/- 9%, n = 4; AAV-PHP.eB n-SIM: 6 +/- 3%, n = 4. AAV5 n-SIM vs. AAV9 n-SIM: p<0.0001; AAV1 n-SIM vs. AAV9 n-SIM: p<0.0001; AAV1 n-SIM vs. AAV9 n-SIM: p=0.0476; AAV-PHP.eB n-SIM vs. AAV9 n-SIM: p<0.0001). AAV1 yielded similar levels to AAV9 in some cases, but with less uniform expression across cortical regions (*Figure 1—figure supplement 1D,E,G*). Finally, the minimally invasive n-SIM procedure resulted in no obvious detrimental effects on animal health, as measured by weight-gain of the n-SIM animals in comparison to non-injected controls (P7-P9 sinus injected at P1: 5 +/- 1 g, n = 8; P7-P9 non-injected: 5 +/- 1 g, n = 5). Sinus injected animals survived for as long as we observed (P63, n = 3), with no evidence of infection or rejection by the dam.

## AAV9 n-SIM at P1 enables efficient labeling of different cell populations in the same brain during the first postnatal week

To demonstrate the compatibility of n-SIM with *in vivo* functional imaging of different cortical neuron populations, we injected 4 µL of AAV9-syn-GCaMP6s at P1 into the transverse sinuses of *Vip*-IRES-Cre/LSL-tdTomato, *Sst*-IRES-Cre/LSL-tdTomato, or *Nkx2.1*-Cre/LSL-tdTomato mice at P1. Transfection with n-SIM results in sufficiently bright GCaMP expression to perform two-photon calcium imaging of layer 2/3 excitatory and inhibitory (Nkx2.1, somatostatin (Sst), Vip) neurons simultaneously as early as P4 (*Figure 2A–I*) and persisting into adulthood. In addition to imaging layer 2/3, labeling through n-SIM also enables imaging of deep cortical layers as early as P9 (*Figure 2H,I*). Time-series traces (*Figure 2C,E,G,I*) show that interneuron subtypes in V1 displayed spontaneous activity as shown by their calcium transients. This indicates that the n-SIM method is suitable for labeling and manipulating gene expression in excitatory and inhibitory neurons across the whole brain during the first postnatal week.

## Simultaneous whole-brain expression of two constructs

In addition to the expression of single transgenes, n-SIM is suitable for whole-brain expression of multiple transgenic constructs simultaneously. For instance, it is possible to express both GCaMP6s and jRCaMP1b across the brain in specific cell types using Cre recombinase in Vip interneurons. To demonstrate this, we co-injected a total volume of 4 µL of AAV9-CAG-flex-GCaMP6s and AAV9-syn-jRCaMP1b (1:1 ratio) into the transverse sinus of *Vip*-IRES-Cre mice at P1 (*Figure 3A*). With this preparation, we observed widespread neuronal expression across the cortical mantle of jRCaMP1b, and Vip interneuron specific expression of GCaMP6s. This expression was suitable, for instance, for macroscopic calcium imaging of Vip interneurons (with GCaMP6) and all neurons (with RCaMP) at P10 (*Figure 3B–E*). This allowed us to visualize and distinguish calcium dynamics from two neuronal populations simultaneously and separately with a single-photon mesoscope in neonates, enabling a

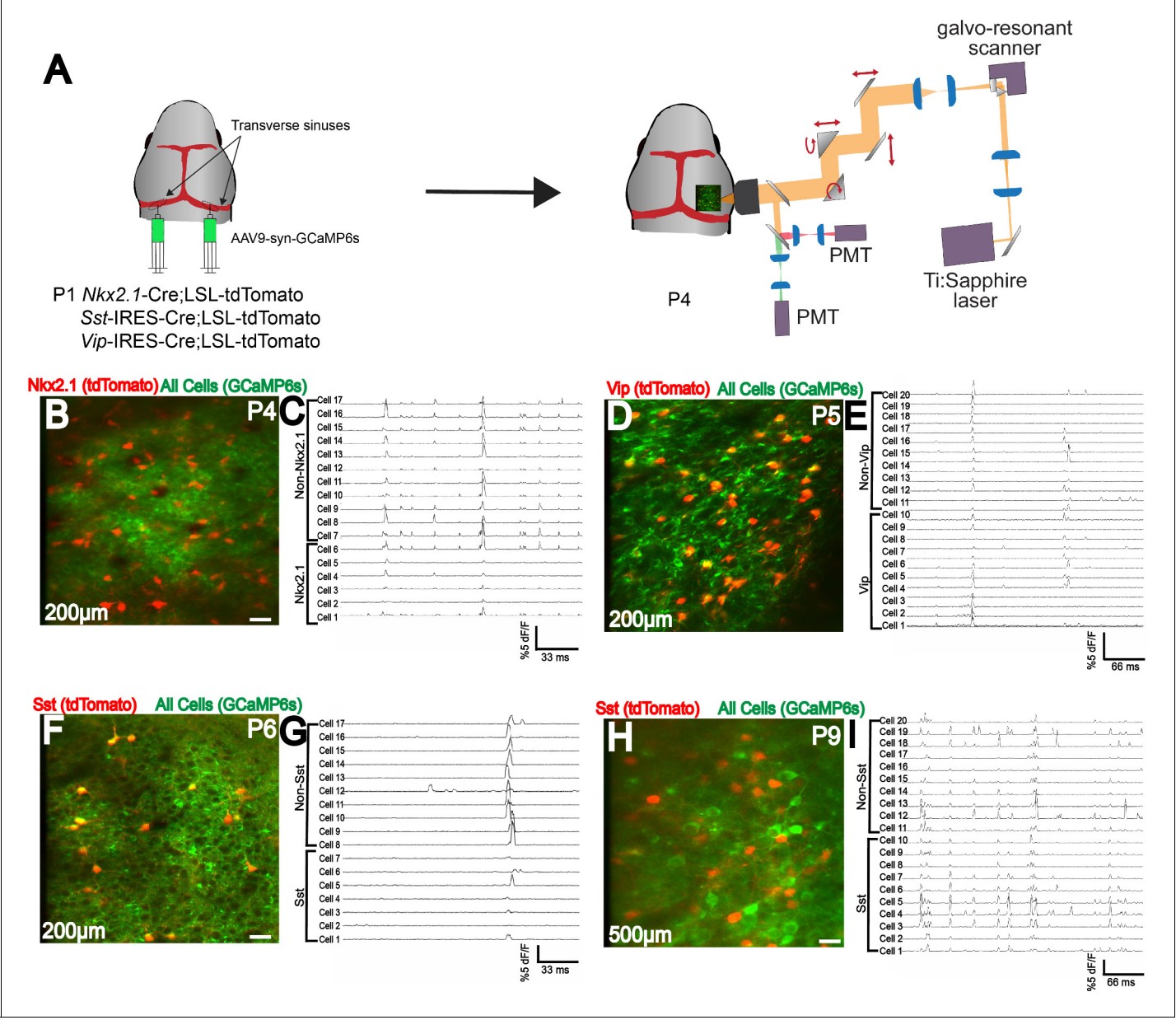

**Figure 2.** Efficient labeling of multiple cell types in neonatal cortex using n-SIM. (**A**) Schematic showing viral injection (AAV9-syn-GCaMP6s) into *Nkx2.1*-Cre;LSL-tdTomato, or *Sst*-IRES-Cre;LSL-tdTomato, or *Vip*-IRES-Cre;LSL-tdTomato to label excitatory and inhibitory neuron populations. 2-photon imaging setup is shown. (**B–I**) Simultaneous two-photon imaging of Nkx2.1, Sst, and Vip interneurons in V1 layers 2/3 along with surrounding pyramidal neurons as early as P4, showing efficient labeling of all interneuron subtypes. Traces adjacent to images show that all interneuron subtypes display calcium transients. (**H,I**) Deep-layer two-photon imaging of Sst interneurons and surrounding pyramidal neurons at P9, showing efficient labeling of deep cortical layers early in development. Note that for illustration purposes, only a subset of neurons have traces displayed in this figure. Scale bar is 20 µm.

direct comparison of the spatiotemporal dynamics of excitatory and inhibitory neurons cortex-wide throughout development without complex interbreeding of various transgenic mouse lines.

This novel methodology makes it possible to study interactions between multiple neuronal and non-neuronal populations simultaneously, while preserving the ability to perform genetic manipulations in specific cell types early in development. In addition, n-SIM enables whole brain expression of various fluorescent neurotransmitter indicators, such as glutamate (iGluSnFR) (*Marvin et al., 2018*) or acetylcholine sensors (GACh) (*Jing et al., 2018*), in order to simultaneously monitor the relationship between neurotransmitter release and cellular activity in neonates (*Figure 3—figure*

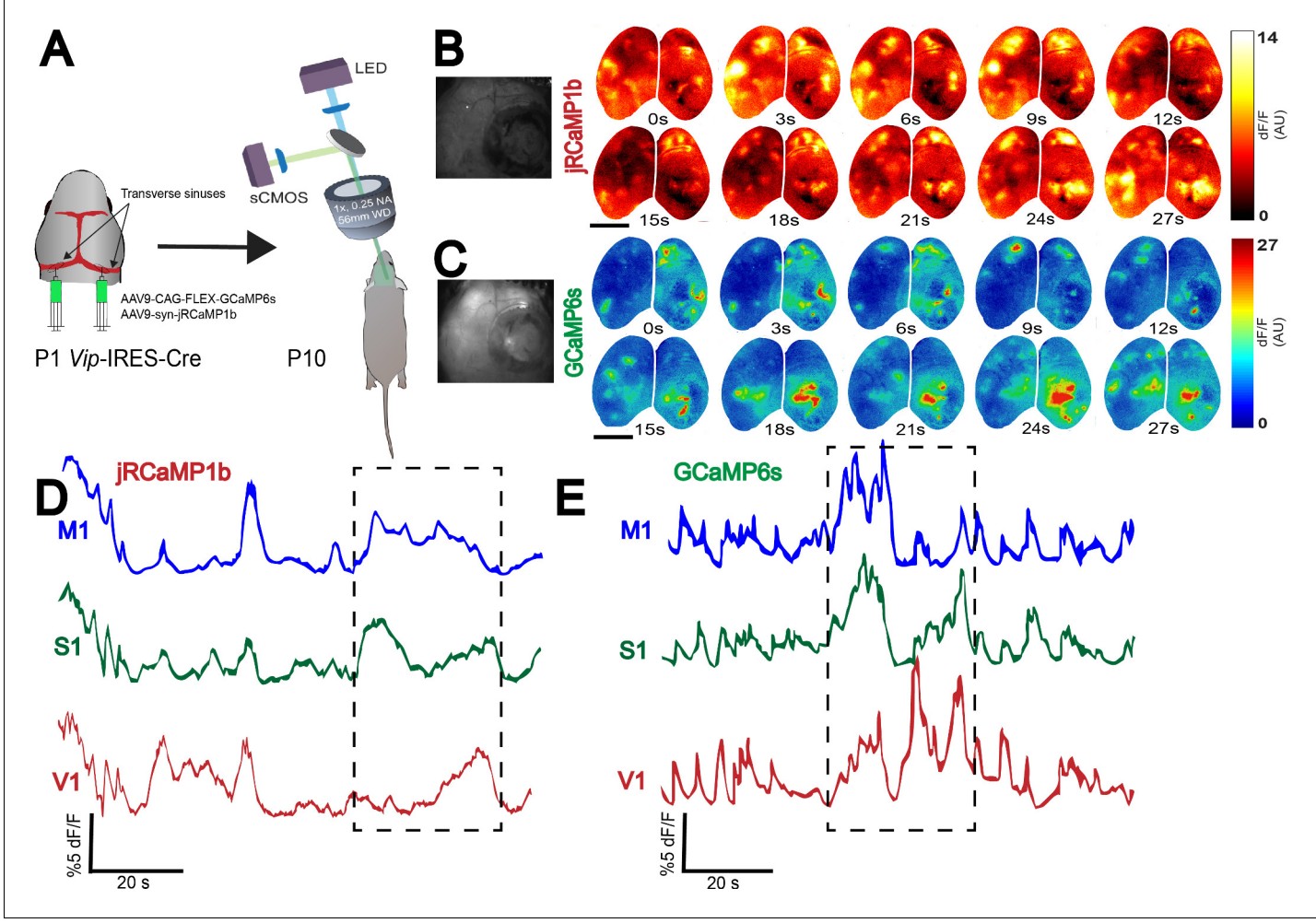

**Figure 3.** Whole-brain expression of multiple viral constructs in neonates. (A) Schematic showing co-injection of two viruses (AAV9-syn-jRCaMP1b and AAV9-CAG-flex-GCaMP6s) into the transverse sinuses of *Vip*-IRES-Cre mice at P0-P1, and widefield imaging of two neural populations in the same brain at P10. Widefield imaging setup is shown in simplified schematic (see Methods section for details). (B,C) Montage of neural activity across cortex imaged using a widefield mesoscope. All neurons are labeled using jRCaMP1b, and only Vip interneurons are labeled using GCaMP6s. Notice that domains of neural activity are observed across the entirety of cortex in both the jRCaMP1b and GCaMP6s channels, demonstrating that functional viral expression is robust across the whole cortex. Photographs on the left represent example frames showing widefield imaging under yellow (B) and blue (C) illumination. (D,E). Traces represent time-series of spontaneous activity measured by calcium transients from motor cortex (M1), somatosensory cortex (S1), and visual cortex (V1). Boxed area of traces in D and E is shown as a montage in (B and C), respectively. Scale bar is 2 μm.

The online version of this article includes the following figure supplement(s) for figure 3:

**Figure supplement 1.** Whole-brain expression of a fluorescent glutamate indicator (iGluSnFr) and pan-neuronal jRCaMP1b via n-SIM.

---

supplement 1). This is especially powerful because deep brain structures such as the basal forebrain are difficult to access for imaging or electrophysiological studies, yet the basal forebrain is known to have widespread cortical projections and is highly involved in modulation of behavioral state and state-dependent cortical processing through Ach release (*Li et al., 2018*). Using n-SIM, we can now study the relationship between activity in these deep brain structures and different cortical regions simultaneously.

## Multi-species compatibility of n-SIM

Another important utility of n-SIM is its compatibility with multiple species, such as rats, that lack the wide array of genetic tools available in mice. We tested n-SIM with P1 injections in Long Evans rats and achieved robust whole-brain expression as early as P6 using 4-8 μL of virus (*Figure 4*). The GCaMP signal had comparable brightness and activity-dependent changes in fluorescence (ΔF/F) as

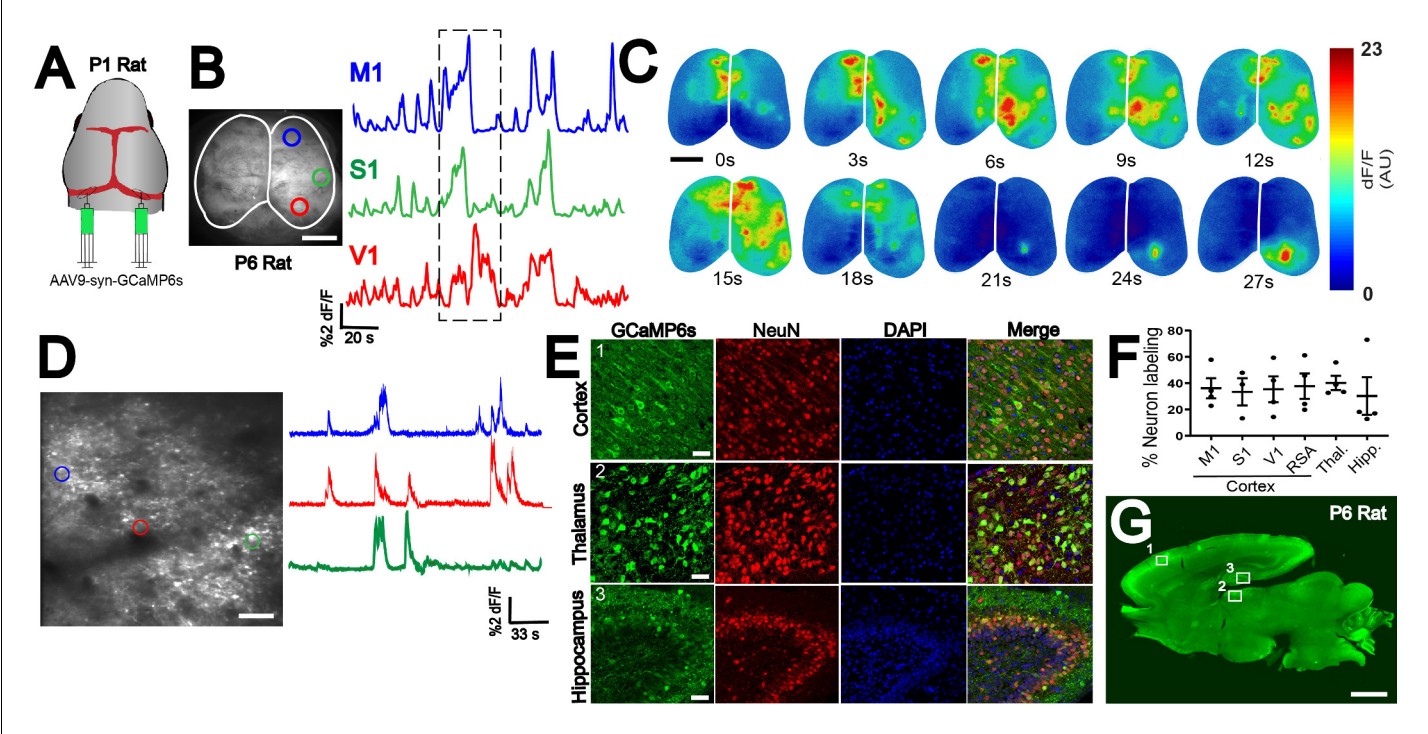

**Figure 4.** n-SIM compatibility in multiple species. (**A**) n-SIM using 4-8 μL of AAV9-syn-GCaMP6s ($1 \times 10^{13}$ vg/mL) at P1 in rats yields whole-brain expression of GCaMP at P6 with strong functional signal across cortex as shown by widefield imaging (**B**). Traces represent time-series of spontaneous activity measured by calcium transients from motor cortex (M1), somatosensory cortex (S1), and visual cortex (V1). (**C**) Boxed area of traces in (**A**) shown in a montage. Scale bar is 4 μm. (**D**) Two-photon imaging of spontaneous cortical activity in V1 from a P6 rat with sample traces from individual neurons marked with color-matched circles. Scale bar is 50 μm. (**E**) Confocal images from boxed areas in F showing dense labeling of neurons in cortex, thalamus, and hippocampus at P6. Scale bar is 50 μm. (**F**) Quantification of neuronal labeling in rats ranging from P6-P9. (**G**) Sagittal sections of a P6 rat brain showing widespread rostral-caudal expression of GCaMP6 across the whole brain. Scale bar is 2 mm. Exposure time: 6000 ms. Minimum-Maximum display range in ImageJ (Unsigned 16-bit range)=73–2500.

The online version of this article includes the following source data for figure 4:

**Source data 1.** Quantification of neuronal labeling in the rat brain.

that observed in mice at a similar age under the same injection conditions (*Figure 4A–C*). Using this method, we were able to detect and quantify spontaneous neural activity during the first week of postnatal development using both widefield (*Figure 4B,C*) and 2-photon calcium imaging (*Figure 4D*) from across the cortex. Immunostaining of sinus injected rats showed robust expression of GCaMP6 across the whole brain as early as P6, as demonstrated by confocal (*Figure 4E*) and epi-fluorescence imaging (*Figure 4G*). Quantification of neuronal labeling using n-SIM shows robust expression across different cortical and subcortical regions in the rat brain (*Figure 4F*; *Figure 4F*–source data 1, M1: 36 +/- 8%, n = 4; S1: 33+/- 10%, n = 3; V1: 35 +/- 10%, n = 4; RSA: 38+/- 10%, n = 4; thalamus: 40 +/- 5%, n = 4; hippocampus: 30 +/- 14%, n = 4). However, it is worth noting that, in our hands, the success rate of n-SIM in rats was approximately 50–60%, whereas in mice the success is 90–100%. This could be due to the thicker skull in rats compared to mice, which can obscure the view of the sinuses during injections. For best results, we recommend injecting the rat pups as early after birth as possible (P0). Increasing the viral titer would also significantly improve the labeling percentage.

## Comparison of n-SIM to ICV

For direct comparison of n-SIM with a previously established method (ICV) for viral vector-driven gene delivery, we performed ICV injections of AAV9-syn-GCaMP6s (4 μL per animal) at P0, and harvested brains at P6, P9, and P16. First, we performed widefield calcium imaging at P9 to assess the quality of functional imaging using this approach (*Figure 5A–C*). We observed that domains of

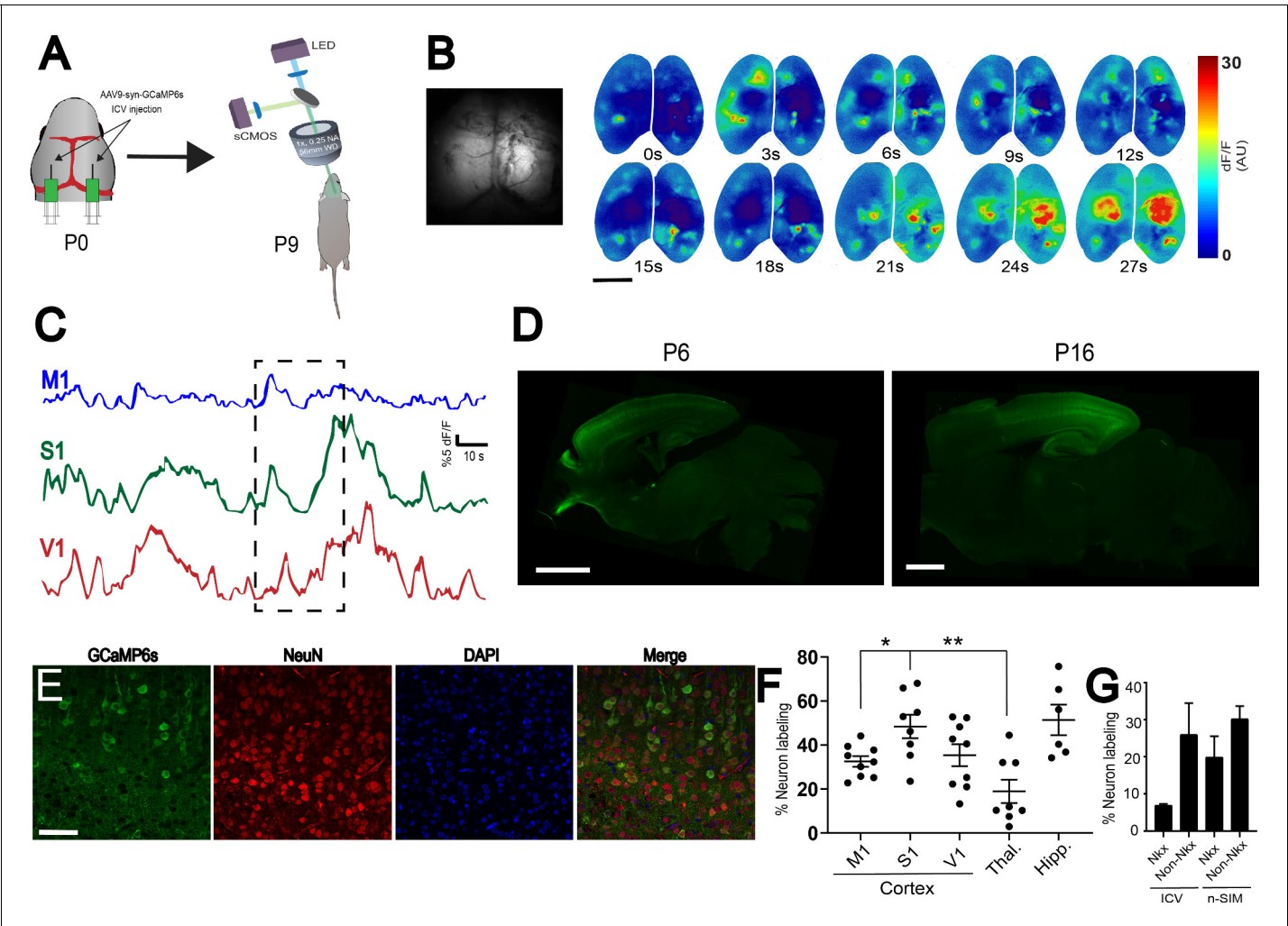

**Figure 5.** Comparison of ICV to n-SIM. (**A**) Schematic showing sites of ICV injection of AAV9-syn-GCaMp6S at P0 and widefield imaging at P9. (**B**) Montage of neural activity across cortex imaged using a widefield mesoscope. Notice that domains of activity are brighter in the caudal parts of cortex relative to the anterior parts. This is supported by the time-series traces (**C**), where the amplitudes of calcium transients from M1 appear to have lower amplitudes than S1 and V1. Photograph on the left represent a frame showing widefield imaging under blue illumination. (**D**) Sagittal sections of P6 and P16 mouse brains with bilateral ICV injections of AAV9-syn-GCaMp6s at P0. P6 Exposure time = 1000 ms. Minimum-Maximum display range in ImageJ (Unsigned 16-bit range)=73–1200. P16 Exposure time = 1000 ms. Minimum-Maximum display range in ImageJ (Unsigned 16-bit range)=0–4500. (**E**) Confocal images from cortex showing neuronal labeling at P9. Scale bar is 50 µm. (**F**) Quantification of overall neuronal labeling from P6, P9, and P16 animals after ICV at P0-P1. Each data point in the plot represents an individual brain. Horizontal lines represent the mean, and vertical lines represent the standard error of the mean. M1 = motor cortex, S1 = somatosensory cortex, V1 = visual cortex, Thal. = thalamus, Hipp. = hippocampus. (**G**) Quantification of Nkx2.1 and non-Nkx2.1 neuronal labeling in mice ranging from P7-P14 injected using ICV or n-SIM at P0-P1. Error bars represent the standard error of the mean.

The online version of this article includes the following source data for figure 5:

**Source data 1.** Quantification of neuronal labeling in the mouse brain after ICV injections.

**Source data 2.** Quantification of Nkx2.1 neuron labeling using n-SIM and ICV.

activity are much brighter in the caudal/middle parts of cortex relative to the anterior parts. This is supported by the time-series traces (*Figure 5C*), where the amplitudes of calcium transients from M1 appear to have lower amplitudes than S1 and V1. Next, we sliced those brains and performed immunohistochemical staining for GCaMP6s and NeuN to quantify neuronal labeling to compare the percentage of excitatory and inhibitory neuronal labeling to n-SIM (*Figure 5D–F*). Overall, we found that ICV labels significantly fewer neurons compared to n-SIM (*Figure 1—figure supplement 1G*, *Figure 1—source data 1*). We also observed that labeling using ICV is obviously non-uniform across

cortical areas, unlike n-SIM (*Figure 5F*, *Figure 5—source data 1*; M1: 32 +/- 2%, n = 9; S1: 48 +/- 5%, n = 8; V1: 35 +/- 5%, n = 10; thalamus: 19 +/- 5%, n = 8; hippocampus: 51 +/- 7%, n = 6). For example, ICV labels significantly fewer neurons in M1 relative to S1 (*Figure 5F*; p=0.0132) and S1 relative to the thalamus (*Figure 5F*; p=0.0016). To test the degree to which infection and GCaMP expression via ICV or n-SIM under the synapsin promotor may be biased to particular cell types, we used *Nkx2.1*-Cre/LSL-tdTomato mice to examine the percentage of interneuron and pyramidal neuron labeling. Nkx2.1 is a marker for all interneurons originating from the medial ganglionic eminence, which account for approximately 70% of all interneurons in the brain (*Butt et al., 2005*). We found that ICV tends to label fewer Nkx2.1-positive interneurons and excitatory neurons compared to n-SIM (*Figure 5G*, *Figure 5—source data 2*; ICV Nkx interneurons labeled: 7 +/- 0%, n = 2; ICV non-Nkx neurons labeled: 26 +/- 8%, n = 2; n-SIM Nkx interneurons labeled: 20 +/- 6%, n = 5; n-SIM non-Nkx neurons labeled: 30 +/- 3%, n = 5).

## Discussion

To our knowledge, AAV9 with n-SIM at P0-P1 is the only method to yield robust and widespread transgene expression in the neonatal brain without reliance on transgenic driver lines. The ability to simply and efficiently label and genetically manipulate brain cells very early in postnatal development provides a ready method to understand how different cell types interact and shape each other's integration into cortical circuits. For instance, little is known about how different interneuron subtypes functionally integrate into circuits in the developing cortex, and sparse interneuron populations are difficult to access early in development. Vip interneurons, for example, account for less than 2% of cortical neurons and are difficult to label, manipulate or examine physiologically during the first postnatal week as they arrive in the cortical plate at P1 but do not finish migrating until P7 (*Miyoshi and Fishell, 2011*). This makes it difficult to study Vip interneurons in relation to surrounding pyramidal or other inhibitory neuron cell types, or to conditionally manipulate interneurons in young neonates, as few of them have arrived at the cortical plate by P0-P1 to transfect through traditional viral injection methods (i.e. cortical injections). n-SIM circumvents the methodological challenges of labeling sparse neuronal populations while preserving the ability to combine functional imaging with genetic labeling, as it transfects migrating interneurons more robustly through blood circulation before they finish migrating and sorting into the cortical plate.

Using n-SIM, it is now easily possible to make inter-species comparisons of development that were previously difficult or inaccessible. For instance, though spontaneous neuronal activity prior to eye opening is thought to instruct cortical development (*Huberman et al., 2006*), it is not clear if common rules govern cortical development prior to the onset of sensation in different species. With n-SIM, we are readily able to examine the patterns of spontaneous cortical activity in developing rats prior to eye opening (~P12), making direct comparisons to similar developmental timepoints in mice.

One major advantage of n-SIM over ICV is that n-SIM circumvents the need to advance a 32-gauge syringe needle through both brain hemispheres to inject the virus, which is associated with massive damage to the cortex and cell death, and potentially elevated immune response which may alter brain development. This is a particular concern for experiments that are done in neonates, when little time may lapse to repair cortical damage. n-SIM offers a far less invasive way to achieve whole-brain expression without damaging the cortex, which is especially vulnerable during this early point in development. In addition, ICV injections involve applying a large amount of pressure on the brain during the process of advancing the syringe needle to penetrate through the skull and brain, which can be damaging to the developing brain. n-SIM circumvents this problem by not applying any pressure on the brain, due to the transverse sinuses being located posterior to the cortex, and the bone tissue above the sinuses is much thinner than the bone tissue above the rest of the brain, thus requiring less pressure to penetrate with a micropipette. In summary, n-SIM provides a way to achieve whole-brain expression without touching the brain and damaging it with a large syringe needle or applying any pressure. Another advantage of n-SIM is the accessibility of the transverse sinuses and the ability to easily visualize the injection site, whereas visual confirmation of the efficacy of ICV injections is very difficult (injections are done 'blind'), and rely on a coordinate system and waiting a period of time to find out whether the injection was successful. Furthermore, ICV coordinates rely on the ability to visualize Lamda through the skin, which can be difficult to do as the skin

becomes thicker after the first postnatal day, resulting in failed injections. We believe that the high success rate (~90%) of n-SIM is attributed to the easy accessibility and the ability to clearly visualize the sinuses. This feature also makes it possible to perform multiple injections into the same sinus to achieve more efficient expression of multiple viral constructs if desired by avoiding mixing before injecting. This approach yields significantly brighter expression for each viral construct in the same brain. Performing multiple injection entries into each hemisphere using ICV would subject the brain parenchyma to greater tissue damage, reducing its feasibility. Furthermore, ICV injections appear to result in patchy and non-uniform expression across cortex in our hands (*Figure 5*) and in previous studies (*Kim et al., 2013*; *McLean et al., 2014*), whereas n-SIM results in more uniform expression across cortex.

### Additional applications of n-SIM

We anticipate that n-SIM will be useful for a wide range of applications, in addition to the experiments described above. For instance, with n-SIM it is possible to edit single as well as multiple genes across the whole neonatal rodent brain *in vivo*, persisting into adulthood, without the time-consuming generation of transgenic animals. n-SIM may also be combined with AAV-mediated CRISPR-Cas9 system (*Swiech et al., 2015*; *Kumar et al., 2018*) to provide a rapid and powerful technology for precise genomic perturbations in specific neuronal subtypes or circuit elements using the Cre-Lox system, thereby enhancing the ability of CRISPR-Cas9 to dissect gene functions in brain processes early in development. n-SIM may also be used to selectivity silence genes in neonates (e.g. via shRNA *Yang et al., 2012*). These examples demonstrate how n-SIM could be used broadly and non-invasively to manipulate cells in health and disease for research or therapeutic purposes.

### Anatomical and functional mapping

We demonstrated how n-SIM can be used to genetically target distinct neural circuits throughout the brain early in development. We specifically show how this approach can be used for functional optical imaging (*Barson et al., 2020*) to study long-range connectivity of individual, genetically defined neurons. n-SIM can similarly be used with optogenetic and chemogenetic applications for dissection of neuronal circuit function. Furthermore, n-SIM could be used for AAV-mediated multi-color labeling, such as Brainbow (*Cai et al., 2013*), for anatomical mapping purposes that require intermingled single cells within the same population to be distinguished from one another. This may be particularly useful for studying cell organization in healthy and diseased brains.

In summary, our results demonstrate a novel method to achieve robust whole-brain expression of transgenes in neonates that enables the labeling and imaging of multiple distinct neuronal populations simultaneously across development. Our method is also applicable for use in different mammalian species, which will aid in unraveling the principles underlying the development of functional organization of the cortex. Our study provides a proof of concept for a powerful tool for widespread gene expression that may also be of significant clinical relevance.

## Materials and methods

**Key resources table**

| Reagent type (species) or resource | Designation | Source or reference | Identifiers | Additional information |
|---|---|---|---|---|
| Genetic reagent (*M. musculus*) | C57BL/6J | The Jackson Laboratory Stock # 000664 | IMSR Cat # JAX: 000664; RRID:IMSR JAX:000664 | |
| Genetic reagent (*M. musculus*) | Ai9(RCL-tdT) | The Jackson Laboratory Stock # 007909 | IMSR Cat# JAX: 007909, RRID:IMSR JAX:007909 | |

*Continued on next page*

*Continued*

| Reagent type (species) or resource | Designation | Source or reference | Identifiers | Additional information |
|---|---|---|---|---|
| Genetic reagent (*M. musculus*) | *Vip*-IRES-cre | The Jackson Laboratory Stock # 010908 | IMSR Cat# JAX:010908, RRID:IMSR JAX:010908 | |
| Genetic reagent (*M. musculus*) | *Nkx2.1*-cre | The Jackson Laboratory Stock # 008661 | IMSR Cat# JAX: 008661, RRID:IMSR JAX:008661 | |
| Genetic reagent (*M. musculus*) | *Sst*-IRES-cre | The Jackson Laboratory Stock # 013044 | IMSR Cat# JAX: 013044, RRID:IMSR JAX: 013044 | |
| Genetic reagent (*R. norvegicus*) | Long Evans rat | Charles River Stock # 006 | Strain Code: 006 | |
| Antibody | anti-GFP (Rabbit monoclonal, Alexa Fluor 488 conjugate) | Thermo Fisher | Thermo Fisher Scientific Cat# A-21311, RRID:AB_221477 | 1:500 |
| Antibody | anti-NeuN (Mouse monoclonal) | Millipore | Millipore Cat# MAB377, RRID:AB_2298772 | 1:500 |
| Antibody | Goat anti-mouse (Mouse polyclonal, Alexa Fluor 555 conjugate) | Thermo Fisher | Thermo Fisher Scientific Cat# A-21422, RRID:AB_2535844 | 1:250 |
| Recombinant DNA reagent | pAAV.Syn. GCaMP6s. WPRE.SV40 (AAV9) | PMID:23868258 | Addgene #100843; RRID:Addgene_100843 | |
| Recombinant DNA reagent | pAAV.CAG. Flex.GCaMP6s. WPRE.SV40 (AAV9) | PMID:23868258 | Addgene # 100842; RRID:Addgene_100842 | |
| Recombinant DNA reagent | pAAV.Syn. NES-jRCaMP1b. WPRE.SV40 (AAV9) | PMID:27011354 | Addgene # 100851; RRID:Addgene_100851 | |
| Recombinant DNA reagent | pAAV.hSyn. iFluSnFr. WPRE.SV40 (AAV9) | PMID:23314171 | Addgene # 98929; RRID:Addgene_98929 | |

All experimental procedures are in accordance with National Institutes of Health guidelines and approved by Yale Institutional Animal Care and Use Committees. Animals are treated in compliance with the U.S. Department of Health and Human Services and Yale University School of Medicine. To validate our method and test its applicability, we describe experiments employing immunohistochemistry and *in vivo* two-photon and widefield calcium imaging in mice and rats we've injected using the n-SIM method, with cross-method validation and control experiments.

## Neonatal transverse sinus injection method (n-SIM)

As described previously briefly (*Barson et al., 2020*), P0-P1 pups were removed from their cages and placed on a warm pad. Each pup was anesthetized on an ice-cold surface for 2–3 min before being transferred to a cooled metal plate. A light microscope was used to visualize the transverse sinuses (located on the dorsal surface of the mouse head, *Figure 1—figure supplement 2A,B*). Next, sterilized fine scissors (Fine Science Tools, CA, USA) were used to make two small cuts (~2 mm) in the skin above each transverse sinus (*Figure 1—figure supplement 2C*).

To inject the virus, we used capillary glass tubes (3.5' #3-000-203-G/X, Drummond Scientific Co, PA, USA), pulled using a P-97 pipette puller (Sutter Instruments, CA, USA) to produce fine tips with

high resistance. The sharp pipettes were filled with mineral oil (M3516, Sigma-Aldrich, NY, USA) then attached to a Nanoject III (Drummond Scientific Co). Most of the mineral oil was ejected using the Nanoject. Next, the vector solution was drawn into the pipette. For accurate movement of the Nanoject-attached-pipette, we used an MP-285 micromanipulator (Sutter Instruments).

The pipette was gently lowered through the skull and into the sinus until the tip of the pipette was observed (using the light microscope) to break through the sinus vessel wall. The pipette tip was retracted until it was 300–400 µm below the surface of the skull, such that the tip resided within the lumen of the sinus. With no delay, 2 or 4 µL of virus was injected at a rate of 20 nL per second. Following a 5 s delay, the pipette was retracted, and the same loading and injection procedure was repeated targeting the opposite hemisphere. The total volume of AAV9-syn-GCaMP6s (Addgene, MA, USA; titer of $1 \times 10^{13}$ vg/mL) virus injected per mouse pup was 4 µL. For RCaMP experiments, we co-injected a total volume of 4 µL of $1 \times 10^{13}$ vg/mL of AAV9-CAG-flex-GCaMP6s (Addgene) and AAV9-syn-jRCaMP1b (Addgene) (1:1 ratio) at P1. For iGluSnFr experiments, we co-injected a total volume of 4 µL of $1 \times 10^{13}$ vg/mL of AAV9-syn-iGluSnFr (Addgene) and AAV9-syn-jRCaMP1b (Addgene) (1:1 ratio) at P1. A successful injection was verified by visualizing viral solution flow in the blood stream evidenced by blanching of the sinus (*Figure 1—figure supplement 2C',C''*). After the injection, the skin was folded back, and a small amount of VetBond glue was applied to the cut. The pup was then placed on a warming pad. After the whole litter was injected, the pups were returned to their home cage and gently rubbed with bedding to prevent rejection by the mother.

## Mice

To label different interneuron subtypes, we used *Nkx2.1*-Cre, *Sst*-IRES-Cre, *Vip*-IRES-Cre, LSL-tdTomato (The Jackson Laboratory (JAX) strains 008661, 013044, 010908, 007909, ME, USA) mice. All mice were housed on a 12 hr light/dark cycle with food and water available ad libitum. For histological validation of sinus injections, we used C57BL6/J mice (JAX 000664). For rat experiments, we used Long-Evans rats (strain code: 006, Charles River, MA, USA).

## Perfusion

After allowing time for expression (4 days minimum), neonatal mice were anesthetized with a Ketamine/Xylazine/Acerpromazine mix (37.5 mg/mL, 1.9 mg/mL, and 0.37 mg/mL, respectively). The anesthetic dose for this combination cocktail was 2.0–3.0 mL/kg administered via intraperitoneal (IP) injection. Next, mice were transcardially perfused with phosphate buffer saline (PBS) at room temperature and then with freshly prepared, ice-cold 4% paraformaldehyde (PFA) solution at 5, 9, 13, or 20 days post-injection. Brains were then removed, immersion fixed in 4% PFA overnight, then rinsed with PBS.

## Comparison of AAV9 n-SIM to other methods

To compare the efficacy of AAV9 n-SIM (AAV9-syn-GCaMP6s) at P1 relative to other AAV serotypes, we injected the same volume (4 µL) of either AAV1-syn-GCaMP6s (titer: $1 \times 10^{13}$ vg/mL, Addgene), AAV5-syn-GCaMP6s (titer: $1 \times 10^{13}$ vg/mL, Addgene), or AAV-PHP.eB-syn-GCaMP6s (titer: $1 \times 10^{13}$ vg/mL, Gradinaru Lab, *Chan et al., 2017*). All injection conditions were the same as described above. To compare n-SIM to temporal vein injections, the latter was performed using the same anesthesia procedure described above. Next, we made a small cut to expose the temporal vein, and we used a Nanoject to inject the virus into the vein. After injecting the temporal veins on each side, the cuts were sealed with Vetbond (Vetbond, 3M, MN, USA) and the pups were returned to their home cage.

For comparison of n-SIM to ICV injections, the latter was performed using the same anesthesia procedure described above. Next, we performed ICV injections as described in *Kim et al., 2014*. Briefly, AAV-syn-GCaMP6s solution was loaded into the Nanoject pipette. Next, the injection sites were identified as 0.8 mm lateral from the sagittal suture, and 1.5 mm from lambda. The pipette was positioned at these coordinates, and then it was advanced through the brain until it reached 1.7 mm below the surface of the skull, then retracted to 1.5 mm before injecting 2 mL of virus solution into the ventricle. Then, the pipette was slowly withdrawn, and the contralateral ventricle was injected using the same procedure (*Kim et al., 2014*). After completing the injections into both hemispheres,

the pups were placed back on the warming pad. The pups were returned to their home cage once their body temperature and skin color returned to normal and the pups began to move.

## Histological processing, immunohistochemistry and imaging

To validate and quantify the levels of expression after n-SIM, neonatal brains were sectioned into 150 µm coronal or sagittal slices using a Leica VT1000S vibratome (Leica, IL, USA). Slices were transferred into 0.04% Triton solution, then blocked overnight with 10% goat serum at 4°C. After blocking, primary antibodies were diluted in the blocking solution (1:500), and slices were incubated in the primary antibody solution for 5 days at 4°C. The primary antibodies used were rabbit anti-GFP conjugated to AF488 (ThermoFisher), and mouse anti-NeuN (Millipore). Next, slices were washed with PBS (3×15 min), and incubated in secondary antibody diluted in blocking solution (1:500) overnight at 4°C. The secondary antibody used was goat anti-mouse AF555 (ThermoFisher). Next, slices were washed with PBS (3×15 min), incubated in anti-DAPI antibody diluted in PBS (1:1000) for 15 min, washed with PBS (3×15 min), and mounted on glass slides using Flouromount G (Vectashield, Vector Laboratory, CA, USA) then cover slipped. For display purposes and assessment of overall brightness, slice images were captured using a Zeiss Apotome microscope (Zeiss, Oberkochen, Germany) with exposure time and contrast settings described in the figure legends. For quantification, slices were imaged using a Zeiss laser scanning confocal microscope (LSM 800) equipped with EC Plan-Neofluar 10×/0.3 (Working Distance = 5.2 mm, Zeiss) and Plan-Apochromat 20×/0.8 (Working Distance = 0.55 mm, Zeiss) objectives to determine co-localization between GCaMP6s+ and NeuN+ signals. Signal quantification was done using ImageJ software. Briefly, images were binarized, regions of interest (ROIs) were selected, and the percent overlap between GCaMP6s+ and NeuN+ cells was manually counted twice by multiple blinded observers to minimize bias.

## *In vivo* imaging

### Surgical preparation

To prepare the animals for functional imaging, mice and rats were anesthetized using 1–2% isoflurane and maintained at 37°C using a water heating pad for the duration of the surgery. The scalp was cleaned with povidone-iodine solution, then topical lidocaine applied and Maloxicam (0.3 mg/kg) administered IP. The skin and fascia layers above the skull were removed to expose the entire dorsal surface of the skull from the posterior edge of the nasal bone to the middle of the interparietal bone, and laterally between the temporal muscles. The skull was thoroughly cleaned with saline, and the edges of the skin incision secured to the skull using Vetbond glue.

To head-fix the animal, we used a custom headpost which consists of two screws (0/80 × 3/16 MS24693-C420 Phillips Flat Head 100° 18/8 Stainless Steel Machine Screws, Mutual Screw and Supply, NJ, USA) placed upside down on the skull (i.e. base first) and secured onto the interparietal and nasal bone with Vetbond, and transparent dental cement (Metabond, Parkell, Inc, NY, USA). To reduce motion and exposure of bone to air, a thin layer of dental cement was applied to all exposed skull. Once the dental cement was dried, it was covered with a thin layer of cyanoacrylate (Maxi-Cure, Bob Smith Industries, CA, USA) to provide a smooth surface for imaging. The head-screws were threaded through the holes of a custom-made metal bar and secured with metal nuts. For two-photon imaging, we made a 3 mm diameter cranial window over visual cortex of the right hemisphere using a dental drill (Ram Products, Inc, NJ, USA), and the cranial window was covered with a double coverslip (Small round cover glass, #1 thickness, 3 mm + Small round cover glass, #1 thickness, 5 mm, Warner Instruments LLC, CT, USA) and sealed using Maxi-Cure. All mice were allowed to recover from head-post surgery for a minimum of three hours before imaging.

## Widefield and two-photon calcium imaging

To validate functional expression of GCaMP across the cortical mantle, widefield calcium imaging was performed using a Zeiss Axiozoom V.16 with PlanNeoFluar Z 1×, 0.25 NA objective. Epifluorescence excitation was performed using a LED source (X-cite TURBO XLED, MA, USA) with 450–495 nm illumination for GCaMP6s and 540–600 nm for jRCaMP1b through a Fitc/Tritc filter cube (Chroma, 59022, VT, USA). Epifluorescence emissions were filtered with Semrock FF01-425/45-25, Semrock FF01-624/40-25, and Semrock FF01-593/LP-25 (Semrock Inc, NY, USA). Emissions were

recorded using a sCMOS camera (pco.edge 4.2) with 512×500 resolution after 4×4 pixel binning, and 100 ms exposure. Images were acquired using Camware software (PCO, MI, USA).

To validate the ability to resolve single cells for functional imaging under the two-photon microscope at different cortical depths, we used a Movable Objective Microscope (MOM) and galvo-resonant scanner (Sutter Instruments). Two-photon excitation was performed using a Ti:Sapphire laser (MaiTai eHP DeepSee, Spectra-Physics, CA, USA) with built-in dispersion compensation. Laser intensity into the microscope was controlled using a Pockels cell (Conoptics, CT, USA) and the laser was expanded with a 1.25 × Galilean beam expander (A254B-100 and A254B+125, Thorlabs, NJ, USA). The laser was focused on the brain using an objective with a 1.7 mm WD and 1 NA (Plan-Apochromat 20x, Zeiss). Fluorescence emissions from the brain were reflected into the emissions path by a FF735Di-02 dichroic mirror (Semrock), filtered with an ET500lp long pass filter (Chroma), and then split by a T565lpxr dichroic mirror (Chroma) into two GaAsP PMTs (H10770PA-40, Hamamatsu, NJ, USA) with ET525/50 m-2p (Chroma) and ET605/70 m-2p (Chroma) filters for detection of green and red emissions, respectively. The two-photon microscope was controlled using ScanImage 2017 (Vidrio Technologies, VA, USA) and images were acquired at 512×512 resolution without bi-directional scanning.

## Calcium imaging data analysis

Two-photon data were motion corrected for x-y displacements by rigid body registration using the moco toolbox (*Dubbs et al., 2016*) in ImageJ (NIH). Motion-corrected frames were tophat filtered across time to compensate for whole frame changes in brightness. ROIs are manually selected, and neuropil signal is removed from each ROI's fluorescence signal as described previously (*Chen et al., 2013*; *Lur et al., 2016*). $\Delta F/F$ was calculated for each cell using the $10^{th}$ percentile as the baseline. For widefield imaging data, $\Delta F/F$ for each pixel was calculated by setting the baseline to the $10^{th}$ percentile value for each pixel across time.

## Statistics

Microsoft Excel 2016 and GraphPad Prism 7.01 (GraphPad Software, CA, USA) were used for data analysis and graph generation. Data are represented in figures as Mean ± SEM. Animal group sizes were chosen based on preliminary data that suggested a large effect size. One animal from each group of mice was excluded from analysis after necropsy due to failed sinus injections. Final mouse group sizes are: AAV9 n-SIM at P1 (n = 10) and P4 (n = 4), AAV1 n-SIM at P1 (n = 4), AAV5 n-SIM at P1 (n = 4), AAV-PHP.eB n-SIM at P1 (n = 4), temporal vein injection at P1 (n = 4), ICV injections at P0-P1 (n = 10). For rat experiments, we excluded 4 animals from analysis due to failed injections (n = 4 rats analyzed). For statistical comparisons across groups, data distributions were found to be normal using the Shapiro-Wilk test, and the parametric t-test (unpaired; two-tailed) was computed using GraphPad Prism. Significance was set at $p < 0.05$ in all cases.

## Acknowledgements

This research was supported by NIH F32 EY028869-01A1, R01 EY015788, R01 EY023105, U01 NS094358, P30 EY026878, R01 MH111424. We thank the family of William Ziegler III for their support. We also thank Yueyi Zhang for help with animal breeding and maintenance, Xinxin Ge for help with data analysis, R Todd Constable, Michael Higley, Jess Cardin and their labs for helpful advice and Daniel Barson for help with the simultaneous 2-photon wide-field imaging microscope. We also thank the Gradinaru lab for the AAV-PHP.eB-syn-GCaMP6s virus, in addition to Drs. Evelyn Lake and Alicja Puscian for helpful comments on the manuscript. The authors declare no competing financial interests.

## Additional information

### Funding

| Funder | Grant reference number | Author |
| --- | --- | --- |
| National Institutes of Health | 1F32EY028869 - 01A1 | Ali S Hamodi |

| National Institutes of Health | R01 EY015788 | Michael C Crair |
|---|---|---|
| National Institutes of Health | R01 EY023105 | Michael C Crair |
| National Institutes of Health | U01 NS094358 | Michael C Crair |
| National Institutes of Health | P30 EY026878 | Michael C Crair |
| National Institutes of Health | R01 MH111424 | Michael C Crair |

The funders had no role in study design, data collection and interpretation, or the decision to submit the work for publication.

### Author contributions

Ali S Hamodi, Conceptualization, Resources, Data curation, Software, Formal analysis, Supervision, Funding acquisition, Validation, Investigation, Visualization, Methodology, Writing - original draft, Project administration, Writing - review and editing; Aude Martinez Sabino, Data curation, Formal analysis, Writing - review and editing; N Dalton Fitzgerald, Dionysia Moschou, Formal analysis, Writing - review and editing, Contributed to quantification of neuronal labeling, and provided comments on the manuscript; Michael C Crair, Conceptualization, Resources, Data curation, Supervision, Funding acquisition, Validation, Investigation, Visualization, Methodology, Project administration

### Author ORCIDs

Ali S Hamodi (ID) https://orcid.org/0000-0002-8398-170X
N Dalton Fitzgerald (ID) http://orcid.org/0000-0001-7794-6898

### Ethics

Animal experimentation: All experimental procedures are in accordance with National Institutes of Health guidelines and approved by Yale Institutional Animal Care and Use Committees (IACUC) protocol (#2017-11141). Animals are treated in compliance with the U.S. Department of Health and Human Services and Yale University School of Medicine. All surgery was performed under isoflurane anesthesia (>P4) or ice anesthesia (<P4)., and every effort made to minimize suffering.

### Decision letter and Author response

Decision letter https://doi.org/10.7554/eLife.53639.sa1
Author response https://doi.org/10.7554/eLife.53639.sa2

## Additional files

### Supplementary files

• Transparent reporting form

### Data availability

Source data files have been provided for Figure 1, Figure 4, and Figure 5.

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
