## [Decision Letter]

Thank you for sending your article entitled "Transverse sinus injections: A novel method for whole-brain vector-driven gene delivery" for peer review at *eLife*. Your article has been evaluated by three peer reviewers, including Sacha Nelson as the Reviewing Editor, and the evaluation has been overseen by Catherine Dulac as the Senior Editor.

The editors and reviewers have consulted regarding your manuscript and have raised an issue that we would like you to clarify prior to making a final decision.

The manuscript makes no mention of prior studies from the Jankowski lab at Baylor utilizing a potentially similar technique: specifically, intraventricular injection of AAV immediately following birth. Prior to the formation of the blood-brain-barrier, this may be functionally quite equivalent to venous injection and indeed at least one of these papers (2013, European J. Neurosci. reports very similar densities of labeled cells (e.g. 88-93% of cortical/hippocampal pyramidal neurons as well as labeled neurons in many other structures) using this approach.

This seems like a more important comparison than the comparison with tail vein injection illustrated in the manuscript. The editors and reviewers would like you to clarify how this prior method relates to that described in this manuscript and in your previously published paper. Please also indicate how you plan to revise the manuscript to make relevant differences clear to readers.

Reviewer #1:

This is likely to be of broad interest to neuroscientists seeking to influence brain gene expression, provided the technique works robustly, and especially to the degree that other related approaches in neonatal animals are less robust or less effective for some applications.

The technique described is a deceptively simple modification of existing viral delivery methods but has the potential to be transformative presuming it works as well as it appears to. The problem of conveniently manipulating gene expression rapidly, persistently and extensively within the mammalian brain remains a major impediment to progress in systems and cellular/molecular neuroscience. The authors, by delivering AAV9 directly into the venous sinuses of the skull before the blood-brain barrier has fully formed (P1-P4) achieve extremely robust, rapid and widespread infection of the cortex, thalamus and midbrain with modest viral loads. During the review discussion, another reviewer pointed out the related work of Jankowski and colleagues who also demonstrated in a series of papers that widespread infection of brain neurons at high rates could be achieved prior to formation of the blood brain barrier. These studies, which are not cited in the present paper, make use of intraventricular injection. But this raises the issue of how different the approach actually is, since before formation of the BBB, blood stream and CSF are potentially in continuity. An additional potentially significant weakness of the paper is that the authors have already described the method (albeit somewhat briefly) in a Nature Methods article this year. The other article is mainly about a dual scale imaging technique. Figure 4A-G of that paper significantly overlaps Figure 1 of this paper. In addition, some of the data in Figure 5B, C of the prior paper overlaps that presented here in Figure 2. Although this reduces the novelty of the present paper, there are significant new results in the new manuscript including: 1) relatively detailed comparison with other existing delivery methods, 2) demonstration of the ability to simultaneously inject and express more than one virus (which greatly increases the utility), 3) a more thorough assessment of the duration of labeling and the required timing of injection, and 4) a demonstration of application to another species (rat).

The technical merit is strong. The authors have done a nice job of examining the distribution of labeled cells across structures and regions but have not addressed any biases that might be present in the cell types infected within a region (e.g. within neocortex).

Figure 3 makes an important point, about the ability to label all cells and a single cell types (via a cre driver) but a key issue is the degree to which expression of the syn-jRCaMP1b may be biased to particular cell types. Figure 1 also uses Syn promoter viruses, but does not address whether the labeled cells are pyramidal and interneurons in the approximately expected ratios or whether infection via the transverse sinus biases the towards one or the other. Simply co-staining for GABA could allow this issue to be addressed quickly and easily.

Reviewer #2:

In this paper, the authors described a method to express AAV in young mice by transverse sinus injection. The paper is clearly presented, and this method appears very useful for developmental studies. However, the authors have recently used this method in a different paper that contains much of the data similar to what's in this paper. It is therefore a major concern whether this manuscript in its current form justifies a new publication.

In a recent publication that they participated in (Barson et al., 2019), the authors used this sinus injection method to express AAV in young mice. The Nature Methods paper is about microscope design, whereas the current one is about virus expression. Thus, a separate paper is useful for readers to find details of the injection method and its effectiveness, etc. However, the current paper does not provide much new information beyond what's already in the other paper. The current Figure 1 is very similar to Figure 4 of the Nature Methods paper, and the Vip cell data in Figure 2 is in Figure 5 of the other paper. In fact, the Nature Methods paper seems to have more quantifications than the current paper.

It is true that the current paper contains new data showing that this method can be used to also express RCaMP (Figure 3) and it can be used in rats (Figure 4). Such new information is useful and necessary. Unfortunately, these data are presented without any quantification. This needs to be corrected.

Additionally, it would be useful to know how long the expression remains stable, possibly for functional studies in adult animals; and whether this method offers an advantage over transgenic mice in that several GCaMP6-expressing transgenic lines seem to display aberrant cortical activity (Steinmetz NA et al., 2017, eNeuro).

Reviewer #3:

The manuscript by Hamodi and colleagues shows a convenient and powerful method for performing widespread transfection of the brain with AAV9 viruses in neonatal mice. The technique is sure to be widely used and cited and in my opinion will be of interest to the broad readership of *eLife*. The authors are exactly right that there are few ways to cause viral expression of proteins in developing animals, and this technique looks like a real advance. The technique is well described and the outcomes are well characterized.

I have only one major suggestion for improvement. The authors are apparently unaware of the work of the Jankowski lab at Baylor, which has developed methods for intracerebroventricular injections in neonatal animals. Paper titles include:

"Viral transduction of the neonatal brain delivers controllable genetic mosaicism for visualizing and manipulating neuronal circuits in vivo".

"Intracerebroventricular Viral Injection of the Neonatal Mouse Brain for Persistent and Widespread Neuronal Transduction".

The authors should compare and contrast their method to the Janowski lab methods, and cite the papers. In addition, the authors may wish to add additional references cited in the Janowski papers that describe other past techniques for injecting.

---

## [Author Response]

The manuscript makes no mention of prior studies from the Jankowski lab at Baylor utilizing a potentially similar technique: specifically, intraventricular injection of AAV immediately following birth. Prior to the formation of the blood-brain-barrier, this may be functionally quite equivalent to venous injection and indeed at least one of these papers (2013, European J. Neurosci. reports very similar densities of labeled cells (e.g. 88-93% of cortical/hippocampal pyramidal neurons as well as labeled neurons in many other structures) using this approach.This seems like a more important comparison than the comparison with tail vein injection illustrated in the manuscript. The editors and reviewers would like you to clarify how this prior method relates to that described in this manuscript and in your previously published paper. Please also indicate how you plan to revise the manuscript to make relevant differences clear to readers.

We thank the reviewers and editors for their helpful comments regarding the manuscript. We have revised the manuscript to address the reviewer’s comments. In particular, we performed additional experiments to directly compare transverse sinus injections (n-SIM) and intracerebroventricular (ICV) injections. We note at the outset that ICV injections require that the injection needle penetrate through the brain (cortex) to reach the ventricle, whereas the transverse sinus injections leave the brain parenchyma intact. Moreover, n-SIM injections result in better transfection and significantly more uniform transfection across the cortex than ICV injections. We also revised the manuscript to address the other points raised by the reviewers (as described below). Again, we thank the reviewers for their insightful comments, which we think we have completely addressed, which has resulted in a significantly improved the manuscript. Thank you for the opportunity to present our revision.

Reviewer #1:This is likely to be of broad interest to neuroscientists seeking to influence brain gene expression, provided the technique works robustly, and especially to the degree that other related approaches in neonatal animals are less robust or less effective for some applications.The technique described is a deceptively simple modification of existing viral delivery methods but has the potential to be transformative presuming it works as well as it appears to. The problem of conveniently manipulating gene expression rapidly, persistently and extensively within the mammalian brain remains a major impediment to progress in systems and cellular/molecular neuroscience. The authors, by delivering AAV9 directly into the venous sinuses of the skull before the blood-brain barrier has fully formed (P1-P4) achieve extremely robust, rapid and widespread infection of the cortex, thalamus and midbrain with modest viral loads. During the review discussion, another reviewer pointed out the related work of Jankowski and colleagues who also demonstrated in a series of papers that widespread infection of brain neurons at high rates could be achieved prior to formation of the blood brain barrier. These studies, which are not cited in the present paper, make use of intraventricular injection. But this raises the issue of how different the approach actually is, since before formation of the BBB, blood stream and CSF are potentially in continuity.

Thank you for the suggestions. We modified the manuscript to include a comparison of n-SIM to the Jankowski lab methods (intracerebroventricular injections, or ICV) and cited those papers, as well as adding additional references cited in the Jankowski papers that describe previous techniques for injecting.

Here are the changes we made to the manuscript upon request by the reviewers:

Added these lines to the Introduction “Intracerebroventricular (ICV) injections were shown to yield strong expression in neonatal brains using only 4μL of virus. However, this method still requires penetrating both brain hemispheres with a 32-gauge needle to inject the virus, which is associated with massive damage to the cortex and cell death, and results in highly non-uniform expression across cortical regions [Kim et al., 2013; McLean et al., 2014; Kim et al., 2014; Passini and Wolfe 2001].”

Added this section to the Results “Comparison of n-SIM to ICV: For direct comparison of n-SIM with a previously established method (ICV) for viral vector-driven gene delivery, we performed ICV injections of AAV9-syn-GCaMP6s (4μL per animal) at P0, and harvested brains at P6, P9, and P16. […] We found that ICV tends to label fewer Nkx2.1-positive interneurons and excitatory neurons compared to n-SIM (Figure 5G, Figure 5—source data 2; ICV Nkx interneurons labeled: 7 +/- 0%, n=2; ICV non-Nkx neurons labeled: 26 +/- 8%, n=2; n-SIM Nkx interneurons labeled: 20 +/- 6%, n=5; n-SIM non-Nkx neurons labeled: 30 +/-3%, n=5).”

Added this section to the Discussion: “One major advantage of n-SIM over ICV is that n-SIM circumvents the need to advance a 32-gauge syringe needle through both brain hemispheres to inject the virus, which is associated with massive damage to the cortex and cell death, and potentially elevated immune response which may alter brain development. […] Furthermore, ICV injections appear to result in patchy and non-uniform expression across cortex in our hands and in previous studies (Kim et al. 2013, Figure 4, Figure 5, Figure 7; McLean et al., 2014, Figure 1), whereas n-SIM results in more uniform expression across cortex.”

Added these references: Kim et al., 2013; McLean et al., 2014; Kim et al., 2014; Passini and Wolfe, 2001.

Added a new figure to show the ICV results “Figure 5”, along with the corresponding figure legend: “Figure 5. Comparison of ICV to n-SIM. […] G. Quantification of Nkx2.1 and non-Nkx2.1 neuronal labeling in mice ranging from P7-P14 injected using ICV or n-SIM at P0-P1. Error bars represent the standard error of the mean.”

Added quantification of overall labeling using ICV to Figure 1—figure supplement 1.

We added a description of how ICV was performed in the Materials and methods section: “For comparison of n-SIM to ICV injections, the latter was performed using the same anesthesia procedure described above. […] The pups were returned to their home cage once their body temperature and skin color returned to normal and the pups began to move.”

An additional potentially significant weakness of the paper is that the authors have already described the method (albeit somewhat briefly) in a Nature Methods article this year. The other article is mainly about a dual scale imaging technique. Figure 4A-G of that paper significantly overlaps Figure 1 of this paper. In addition, some of the data in Figure 5B, C of the prior paper overlaps that presented here in Figure 2. Although this reduces the novelty of the present paper, there are significant new results in the new manuscript including: 1) relatively detailed comparison with other existing delivery methods, 2) demonstration of the ability to simultaneously inject and express more than one virus (which greatly increases the utility), 3) a more thorough assessment of the duration of labeling and the required timing of injection, and 4) a demonstration of application to another species (rat).The technical merit is strong. The authors have done a nice job of examining the distribution of labeled cells across structures and regions but have not addressed any biases that might be present in the cell types infected within a region (e.g. within neocortex).Figure 3 makes an important point, about the ability to label all cells and a single cell types (via a cre driver) but a key issue is the degree to which expression of the syn-jRCaMP1b may be biased to particular cell types. Figure 1 also uses Syn promoter viruses, but does not address whether the labeled cells are pyramidal and interneurons in the approximately expected ratios or whether infection via the transverse sinus biases the towards one or the other. Simply co-staining for GABA could allow this issue to be addressed quickly and easily.

Thank you for the suggestions. We have addressed potential biases present in cell types infected within cortex by performing n-SIM of AAV9-syn-GCaMP6s into Nkx2.1-cre/flox-TdTomato to examine the percentage of interneuron and pyramidal neuron labeling. Nkx2.1 is a marker for all interneurons originating from the medial ganglionic eminence, which account for approximately 70% of all interneurons in the brain. We believe this is a more technically convenient approach than staining for GABA and is sufficient to address this point. We found that n-SIM labels 20 +/- 6% of Nkx2.1 cortical interneurons (n=5 animals), and 30 +/- 3% of non-Nkx2.1 cortical neurons (n=5 animals). We made the following changes to include these new results:

We added the description of this result in the “Comparison of n-SIM to ICV” section: “To test the degree to which infection and GCaMP expression via ICV or n-SIM under the synapsin promotor may be biased to particular cell types, we used Nkx2.1-cre/flox-TdTomato mice to examine the percentage of interneuron and pyramidal neuron labeling. Nkx2.1 is a marker for all interneurons originating from the medial ganglionic eminence, which account for approximately 70% of all interneurons in the brain [Butt et al., 2005]. We found that ICV tends to label fewer Nkx2.1-positive interneurons and excitatory neurons compared to n-SIM (Figure 5G, Figure 5—source data 2; ICV Nkx interneurons labeled: 7 +/- 0%, n=2; ICV non-Nkx neurons labeled: 26 +/- 8%, n=2; n-SIM Nkx interneurons labeled: 20 +/- 6%, n=5; n-SIM non-Nkx neurons labeled: 30 +/-3%, n=5).”

We added the results of the Nkx2.1-cre/flox-TdTomato mice experiments in panel G of Figure 5, along with the figure legend description of this part of the figure: “G. Quantification of Nkx2.1 and non-Nkx2.1 neuronal labeling in mice ranging from P7-P14 injected using ICV or n-SIM at P0-P1. Error bars represent the standard error of the mean.”

Reviewer #2:In this paper, the authors described a method to express AAV in young mice by transverse sinus injection. The paper is clearly presented, and this method appears very useful for developmental studies. However, the authors have recently used this method in a different paper that contains much of the data similar to what's in this paper. It is therefore a major concern whether this manuscript in its current form justifies a new publication.In a recent publication that they participated in (Barson et al., 2019), the authors used this sinus injection method to express AAV in young mice. The Nature Methods paper is about microscope design, whereas the current one is about virus expression. Thus, a separate paper is useful for readers to find details of the injection method and its effectiveness, etc. However, the current paper does not provide much new information beyond what's already in the other paper. The current Figure 1 is very similar to Figure 4 of the Nature Methods paper, and the Vip cell data in Figure 2 is in Figure 5 of the other paper. In fact, the Nature Methods paper seems to have more quantifications than the current paper.

We agree that there is some overlap in Figure 1 of this paper with Figure 4 of the other paper. In our opinion, the first figure of this paper is necessary to bring the method into context. However, there is a substantial amount of information in Figure 1 of this paper that was not present in Figure 4 of the previous paper that both substantiates and expands the wide application of the n-SIM technique. In particular: 1) Description of widespread expression across the whole brain as early as P5 (Panels B, C) with our method, ‘n-SIM’. 2) The role of timing of injections (P1 vs. P4) in the efficiency of expression (Panel F). 3) AAV9 yields vastly superior expression than AAV-PHPeB (Panel G). 4) Description and quantification of %neuronal labeling in cortex and thalamus at P6, P9 after P1 injection, and P14 after P4 injection (Panel H). 5) Quantification of %neuronal labeling in a variety of cortical and subcortical brain regions (I). 6) AAV delivery via n-SIM yields robust expression across all cortical layers (Panel N). 7) Sample images showing robust expression in hippocampus (Panels T-W).

In our recent paper in Nature Methods, we used n-SIM to express AAV in *juvenile* mice (>P16) and achieve pan-neuronal expression of GCaMP6s, allowing us to image calcium activity from Vip and excitatory neurons simultaneously. However, in our current paper, we describe the ability of n-SIM to achieve pan-neuronal expression in *neonatal* mice (P4), and we show the ability to achieve robust expression of AAV in interneurons during the first postnatal week, which is challenging to do as interneurons do not finish migrating into cortex until P7, thus making it difficult to label/manipulate them using traditional approaches (i.e. cortical injections). Therefore, the Vip cell data in Figure 2 of this paper does not overlap with Figure 5 of the previous paper. In addition, we show that all interneuron subtypes are labeled during the first postnatal week using n-SIM, and the expression is sufficiently bright to enable 2-photon imaging of deep cortical layers, in addition to superficial layers.

It is true that the current paper contains new data showing that this method can be used to also express RCaMP (Figure 3) and it can be used in rats (Figure 4). Such new information is useful and necessary. Unfortunately, these data are presented without any quantification. This needs to be corrected.

Thank you for the suggestion. We agree that these data in Figure 3 and Figure 4 need to be presented with quantification, and we have addressed this issue in the revised manuscript.

Figure 3: As discussed above, we addressed biases present in cell types infected within cortex by performing n-SIM of AAV9-syn-GCaMP6s into Nkx2.1-cre/flox-TdTomato to examine the percentage of interneuron and pyramidal neuron labeling. Nkx2.1 is a marker for all interneurons originating from the medial ganglionic eminence, which accounts for approximately 70% of all interneurons in the brain.

Figure 4: We quantified the percentage of neuronal labeling in rats. We achieved this by performing additional experiments using n-SIM AAV9-syn-GCaMP6s at P1 in Long-Evans rats and harvesting their brains at P6 and P14. Next, we sliced those brains and performed immunohistochemical staining for GCaMP6s and NeuN to quantify the percentage of neuronal labeling in the rat brain with n-SIM. We added a new panel to show these results in Figure 4F, along with the following description of these results in the “Multispecies compatibility of n-SIM” section:

“Quantification of neuronal labeling using n-SIM shows robust expression across different cortical and subcortical regions in the rat brain (Figure 4F; Figure 4—source data 1F, M1: 36 +/- 8%, n=4; S1: 33 +/- 10%, n=3; V1: 35 +/- 10%, n=4; RSA: 38 +/- 10%, n=4; thalamus: 40 +/- 5%, n=4; hippocampus: 30 +/- 14%, n=4). […] Furthermore, increasing the viral titer would significantly improve the labeling percentage.”

We updated the Figure 4 legend to describe panel F.

Additionally, it would be useful to know how long the expression remains stable, possibly for functional studies in adult animals.

To address this point, we quantified expression in adult mice (P45) after performing n-SIM at P0-P1. We found that expression remains robust in adult (P45) mice. As mentioned above, these results have now been added to Figure 1.

And whether this method offers an advantage over transgenic mice in that several GCaMP6-expressing transgenic lines seem to display aberrant cortical activity (Steinmetz NA et al., 2017, eNeuro).

One of the advantages of using n-SIM rather than relying on GCaMP6s-expressing transgenic lines is that sinus-injected animals are completely healthy and exhibit normal body weight and growth relative to control animals. In our recent paper (Barson et al., 2019), we performed electrophysiological recordings (Supplementary Figure 8) showing that sinus-injected animals do not display aberrant cortical activity as shown by local field potential recordings and mean firing rate of regular spiking neurons.

Reviewer #3:[…] I have only one major suggestion for improvement. The authors are apparently unaware of the work of the Jankowski lab at Baylor, which has developed methods for intracerebroventricular injections in neonatal animals. Paper titles include:"Viral transduction of the neonatal brain delivers controllable genetic mosaicism for visualizing and manipulating neuronal circuits in vivo"."Intracerebroventricular Viral Injection of the Neonatal Mouse Brain for Persistent and Widespread Neuronal Transduction".The authors should compare and contrast their method to the Janowski lab methods, and cite the papers. In addition, the authors may wish to add additional references cited in the Janowski papers that describe other past techniques for injecting.

Thank you for this suggestion, which is similar to that noted by the other reviewers. As described above in our response to reviewer 1, we modified the manuscript to include a comparison of n-SIM to the Jankowski lab methods (intracerebroventricular injections, or ICV) and cited those papers, as well as adding additional references cited in the Jankowski papers that describe previous techniques for injecting.

Finally, we came to realize that we uploaded an outdated version of Figure 1—figure supplement 2 during the original submission. This has been corrected.